# Structural basis for activation of fluorogenic dyes by an RNA aptamer lacking a G-quadruplex motif

Sandip A. Shelke[1], Yaming Shao[1], Artur Laski[1], Deepak Koirala[1], Benjamin P. Weissman[2], James R. Fuller [1], Xiaohong Tan[3,4], Tudor P. Constantin[4], Alan S. Waggoner [5,6], Marcel P. Bruchez [4,5,6], Bruce A. Armitage[3,4] & Joseph A. Piccirilli [1,2]

The DIR2s RNA aptamer, a second-generation, in-vitro selected binder to dimethylindole red (DIR), activates the fluorescence of cyanine dyes, DIR and oxazole thiazole blue (OTB), allowing detection of two well-resolved emission colors. Using Fab BL3-6 and its cognate hairpin as a crystallization module, we solved the crystal structures of both the apo and OTB-$SO_3$ bound forms of DIR2s at 2.0 Å and 1.8 Å resolution, respectively. DIR2s adopts a compact, tuning fork-like architecture comprised of a helix and two short stem-loops oriented in parallel to create the ligand binding site through tertiary interactions. The OTB-$SO_3$ fluorophore binds in a planar conformation to a claw-like structure formed by a purine base-triple, which provides a stacking platform for OTB-$SO_3$, and an unpaired nucleotide, which partially caps the binding site from the top. The absence of a G-quartet or base tetrad makes the DIR2s aptamer unique among fluorogenic RNAs with known 3D structure.

[1] Department of Biochemistry and Molecular Biology, The University of Chicago, Chicago, IL 60637, USA. [2] Department of Chemistry, The University of Chicago, Chicago, IL 60637, USA. [3] Center for Nucleic Acids Science and Technology, Carnegie Mellon University, Pittsburgh, PA 15213, USA. [4] Department of Chemistry, Carnegie Mellon University, Pittsburgh, PA 15213, USA. [5] Department of Biological Sciences, Carnegie Mellon University, Pittsburgh, PA 15213, USA. [6] Molecular Biosensor and Imaging Center, Carnegie Mellon University, Pittsburgh, PA 15213, USA. Correspondence and requests for materials should be addressed to J.A.P. (email: jpicciri@uchicago.edu)

RNA-based fluoromodules have become a valuable tool in biomedical research for RNA sensing and imaging applications[1]. They consist of an RNA aptamer that can fold into a three-dimensional structure, recognize a profluorescent small molecule, and activate its fluorescence[2,3]. Although RNA-based fluoromodule development remains less advanced compared with protein-based fluoromodules[4] due to the lack of inherently fluorescent RNAs, recently discovered fluorescent RNA aptamers have facilitated cellular imaging, localization, and tracking of RNAs in real time[1,5]. In vitro selection from random RNA populations has given rise to numerous RNA aptamers that bind and activate the fluorescence of small-molecule chromophores such as flavin mononucleotide[6,7], sulforhodamine B[8–10], malachite green (MG)[2], derivatives of Hoechst 33258[11], black hole quencher (BHQ)-conjugated fluorophores[12,13], dimethylindole red (DIR)[3], dinitroaniline-conjugated dyes[14], and small-molecule analogs based on the chromophore in fluorescent proteins[15,16]. RNA aptamers such as Spinach[15], Spinach2[17], Broccoli[18], and Corn[16], which activate fluorescence of 3,5-difluoro-4-hydroxybenzylidene imidazolinone and 3,5-difluoro-4-hydroxybenzylidene imidazolinone-2-oxime (DFHO), (small-molecule fluorophore analogs of green and red fluorescent proteins, respectively) and RNA Mango[19,20], which activates fluorescence of thiazole orange derivatives, have been used for RNA imaging studies both in vitro and in cellulo[15,17,21,22]. These aptamers can be encoded genetically into RNAs of interest and upon incubation with their cognate fluorophores allowed to study RNAs in a manner analogous to the use of green fluorescent proteins[23] to study proteins[4]. For example, Spinach has been used as an RNA visualization probe in bacterial and mammalian cells[15,17,21], as biosensors for detection of metal ions[24], cellular metabolites[25–27], and proteins[28], and for construction of molecular beacons[29]. Nevertheless, existing fluoromodules remain constrained by limitations, including inadequate fluorophore-binding affinity, misfolding tendencies, and loss of fluorescence through photobleaching[17,30,31]. Advanced generations of these fluoromodules partly overcome these limitations but their use for in vivo applications remains problematic. Another potential limitation concerns the presence of an essential G-quartet motif[32,33] in the Spinach[34,35], Corn[36], and Mango[37] RNA fluoromodules as eukaryotic cells have

machinery that specifically inhibits the formation of G-quartet folded structures[38]. It is therefore important to continue investigation of other RNA aptamers that bind and activate small-molecule fluorescence both to develop a framework for relating their structure to their functional properties and to identify motifs that can function without a quadruplex motif. For example, a recently developed RNA imaging platform for tracking mRNA and small noncoding RNAs uses the cobalamin riboswitch as an RNA tag and fluorescent dye fused to cobalamin as fluorescence quencher that exhibits fluorescence turn-on upon binding to the riboswitch[39].

Armitage and colleagues generated an RNA fluoromodule against the fluorogenic cyanine dye, DIR[3,40]. First, to reduce the background fluorescence of the dye in an intracellular environment, a prerequisite for fluoromodule-based imaging applications, they precluded nonspecific intercalation by the dye through the incorporation of a geminal dimethyl group on the indole ring and a negatively charged anionic sulfonate substituent on the quinoline ring, which introduces repulsive electrostatics to the polyanionic backbone of RNA (Fig. 1a). Next, they performed in vitro selection for sequences that bind DIR from an RNA library[41,42] containing a constant stem-loop forming sequence flanked by asymmetric randomized regions[40]. DIR2s (DIR second selection) RNA aptamer obtained represents the second generation of DIR binding RNA aptamers that bind to DIR dye and activate its red fluorescence. The aptamer exhibits promiscuity with respect to dye recognition, binding with nanomolar affinity to red fluorescence emitting DIR, as well as blue fluorescence emitting oxazole thiazole blue (OTB)[5] dyes and their derivatives. Tan et al. used this property to monitor the cell surface expression and internalization of the epidermal growth factor receptor (EGFR) by fusing a known EGFR-binding aptamer to the DIR2s aptamer and imaging at different time points using either red (DIR) or blue (OTB) dyes[40]. Also, both dyes have shown excellent photostability when complexed with the DIR2s RNA aptamer compared with the MG aptamer[40].

To understand the promiscuity of the DIR2s aptamer and its ability to bind DIR and OTB dyes, we sought to obtain the crystal structure of the aptamer in complex with fluorogens. We used the Fab BL3-6 antibody as an RNA crystallization chaperone[43]. Originally selected against the class I ligase ribozyme[44,45], Fab

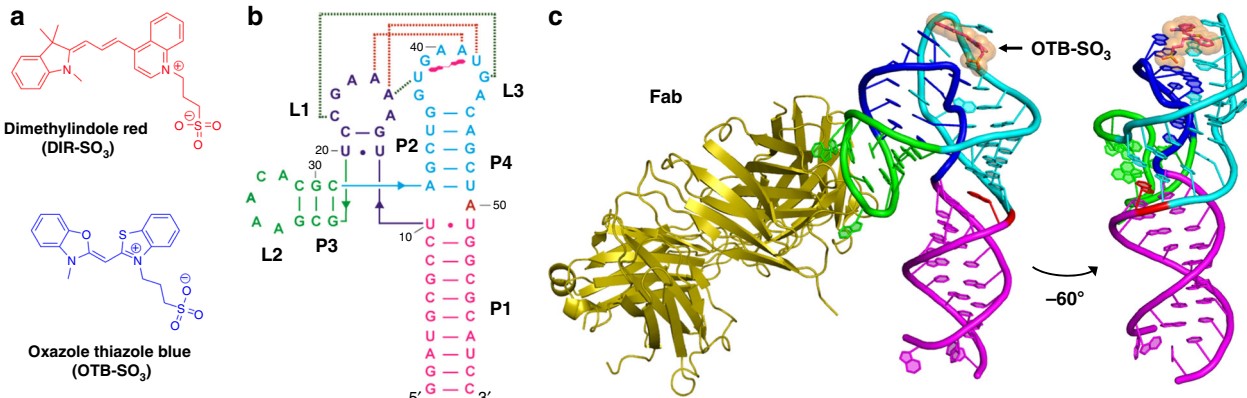

**Fig. 1** Overall structure of OTB-SO3 fluorophore-bound DIR2s aptamer-Fab complex. **a** Chemical structure of dimethylindole red (DIR) and oxazole thiazole blue (OTB-SO3) fluorophore ligands. **b** The secondary structure of the DIR2s aptamer•OTB–SO3 complex derived from the crystal structure. The dotted lines represent kissing loop interactions between loops L1 and L3, where green and red dotted lines indicate Watson–Crick (WC) and noncanonical interactions, respectively. Within stem regions, the WC and non-WC base pairs are denoted by dashes and dots, respectively; lines containing arrowheads denote the connectivity in RNA strand. **c** Cartoon representation of the crystal structure of the DIR2s aptamer•OTB-SO3•Fab BL3-6 ternary complex and a 60° rotated view (for clarity, Fab is not shown in 60° rotated view). The OTB-SO3 fluorophore is represented by sticks and transparent spheres at the apex of the loops L1 and L3. Sequence G23AAACAC29 is the Fab BL3-6 binding tag (loop L2) grafted into the parent aptamer (Supplementary Fig. 1). PDB ID: 6DB8

BL3-6 binds to a stem-loop structure containing an AAACA pentaloop closed by a G-C base pair, which can be grafted into target RNA constructs to create a binding site for the Fab[34]. We used this strategy to crystallize and solve the crystal structures of DIR2s RNA aptamer in the absence of ligand and co-crystallized with OTB-SO$_3$ dye at 2.0 Å and 1.8 Å resolution, respectively. Our structures reveal that the DIR2s aptamer adopts a tuning fork-like fold consisting of two short stem-loops and a proximal helix. Tertiary interactions between the two loops create a stacking platform for the fluorophore ligand formed by three purine nucleobases, including a guanine nucleobase that forms hydrogen bonds to the ligand's –SO$_3$ moiety. From above, only a single adenosine residue at the apex of the aptamer sandwiches the fluorophore, likely accounting for the relaxed specificity.

## Results

**Fab antibody-assisted RNA crystallography**. We replaced the four-nucleotide loop (UUCG) from the constant stem-loop domain of the wild-type (WT) DIR2s aptamer with a 7-nucleotide GAAACAC motif to create a binding site for the chaperone Fab BL3-6 (Supplementary Fig. 1a). Fab BL3-6 binds to the modified DIR2s aptamer with an affinity ($K_d = 67 \pm 20$ nM; error reports SD from three independent measurements) similar to that of binding affinity reported for either Spinach RNA grafted with GAAACAC motif[34] or isolated hairpin from class I ligase RNA[43]. The fluorescence activation of DIR and OTB-SO$_3$ dyes complexed with the DIR2s aptamer remained unaffected by the hairpin graft or binding of the Fab compared with the parent aptamer ($K_d = 708 \pm 16$ nM; error reports SD from three independent measurements), indicating that the BL3-6 binding loop had no adverse effects on its overall folding (Supplementary Fig. 1b). DIR2s-OTB-SO$_3$ RNA-fluorophore complex was co-crystallized with Fab BL3-6, and the structure was solved at 1.8 Å resolution (Fig. 1b, c) with one Fab–RNA-fluorophore complex molecule per crystallographic asymmetric unit (Supplementary Note 1; Supplementary Fig. 2, 3 and 4). After model building and refinement, the final $R_{free}$ and $R_{work}$ values were 0.21 and 0.24, respectively (Table 1). The initial phases were obtained by molecular replacement using Fab-BL3-6 coordinates (PDB: 3IVK) as a search model[43]. The 1.8 Å-resolution electron density map allowed unambiguous tracing of the RNA and revealed a fluorophore-binding site with a clear non-nucleotide density corresponding to the fluorophore. The refined structural factors clearly showed electron density corresponding to the two rings of the OTB dye in planar configuration between the two layers formed by A40 and A41-G39 (Supplementary Fig. 5, 15 and 16). Although we observed no electron density corresponding to the propyl side chain of the OTB-SO$_3$, we did observe clear electron density for the sulfonate group (Supplementary Fig. 6). Nucleotides from the two-base pairs at the base of the stem P1 could not be modeled accurately due to the poor electron density.

**Overall structure of DIR2s–OTB–SO$_3$ complex**. The 60-mer RNA aptamer construct adopts an elongated structure comprised of four helices P1, P2, P3, and P4, and three loops L1, L3, and the Fab BL3-6 binding loop L2 (Fig. 1). The stem-loop P4-L3 co-axially stacks on the helix P1 while loop L1 and its associated short stem P2 orients parallel to and interacts with the loop L3 through base pairing and tertiary interactions. Overall, stem-loops P2-L1 and P4-L3 adopt a tuning fork-like fold, with the stem P1 forming the handle and stem-loops P2-L1 and P4-L3 forming the prongs stabilized by extensive interactions between loops L1 and L3 (Fig. 2a). Stem-loop P3-L2, containing the grafted, Fab-binding L2 pentaloop, projects away from the P1–P4 helical axis at an angle of ≈ 100°, directing the Fab away from the

### Table 1 Data collection and refinement statistics for the DIR2s aptamer apo and OTB-SO$_3$ bound forms

|  | Apo | OTB-SO$_3$ bound |
|---|---|---|
| **Data collection** | | |
| Space group | P2$_1$ 2$_1$ 2$_1$ | C2 2 2$_1$ |
| Resolution (Å) | 83.81–2.03 | 80.27–1.86 |
|  | (2.08–2.03)[a] | (1.91–1.86)[a] |
| Cell dimensions | | |
| a, b, c (Å) | 83.78, 83.81, 109.93 | 109.16, 118.44, 119.31 |
| α, b, γ (°) | 90, 90, 90 | 90, 90, 90 |
| $R_{merge}$ (%) | 4 (170) | 6 (230) |
| $I/\sigma I$ | 20.1 (0.9) | 21.6 (0.9) |
| Completeness (%) | 99.3 (96.9) | 99.5 (94.9) |
| Redundancy | 5.1 (4.9) | 12.9 (11.5) |
| Refinement | | |
| No. reflections | 50,521 | 63,985 |
| $R_{work}/R_{free}$ | 0.230/0.257 | 0.2193/0.2331 |
| R.M.S deviations | | |
| Bond angles (°) | 0.756 | 0.740 |
| Bond length (Å) | 0.003 | 0.004 |
| Ramachandran plot of protein residues | | |
| Preferred regions (%) | 95.23 | 96.74 |
| Allowed regions (%) | 4.3 | 3.02 |
| Disallowed region (%) | 0.48 | 0.23 |
| B-factors | 63.0 | 52.0 |

[a]Values in the parentheses are for the highest resolution shell

RNA core. The interactions between loops L1 and L3 create a ligand-binding pocket that engages the fluorophore OTB-SO$_3$ in a stacking sandwich with the bottom layer formed by three purine bases and the top layer consisting of a single-adenine nucleobase.

**Loop–loop interactions**. The L1 and L3 loops containing six and eight nucleotides, respectively, are anchored together through four base pairing interactions (A14-U38, A14•U42, C18-G43, and A15•A41), including a U-A-U base triple and stacking interactions. Two of these base pairs, A14-U38 and C18-G43, form canonical Watson–Crick (WC) pairings, whereas the other two (A14•U42 and A15•A41) form noncanonical[46] pairing interactions (Fig. 2a). Among the noncanonical base pairs, trans A15•A41 forms via Hoogsteen faces interactions, as commonly observed in rRNA[47] (Fig. 2b). Including the two hydrogen bonds to A15, the A41 nucleobase engages in five hydrogen-bonding interactions. The exocyclic amino group interacts with the 2′-OH of A14, and N1 and N3 interact with the exocyclic –NH$_2$ and 2′-OH groups of G39, respectively (Fig. 2b). A14 of loop L1 forms a base triple with U38 (trans WC-WC) and U42 (trans Hoogsteen-WC) from loop L3 (Fig. 2c) and forms stacking interactions with the fluorophore-binding nucleotide G39 (Fig. 2a). Below these three consecutive layers of stacking interactions, the unpaired A13 from L1 protrudes into the L3 loop and interacts with G43 through nucleobase stacking and hydrogen bond donation from both its exocyclic amino group to the G43 ribose 5′- and 4′-oxygens and its 2′-OH to N7 of G43 (Fig. 2d). In addition, A13 N7 accepts a hydrogen bond from a water molecule, which in turn forms hydrogen bonds to 4′-oxygen and 2′-OH of G43 and A44, respectively. A13 induces a kink in the phosphate backbone of the RNA, which places the A13 phosphate nonbridging oxygen (pro-S) near the G37 and C45 of loop L3, within hydrogen-bonding distance of G37 N$^1$H and the C45 exocyclic amine (Fig. 2d). G12, which forms the closing base pair of the L1 loop with C19, supports the L1–L3 interaction through direct and water-mediated hydrogen bonds from its minor groove face to L3 (G43) and C45 (the P4 closing base pair; Fig. 2e). In the loop L3,

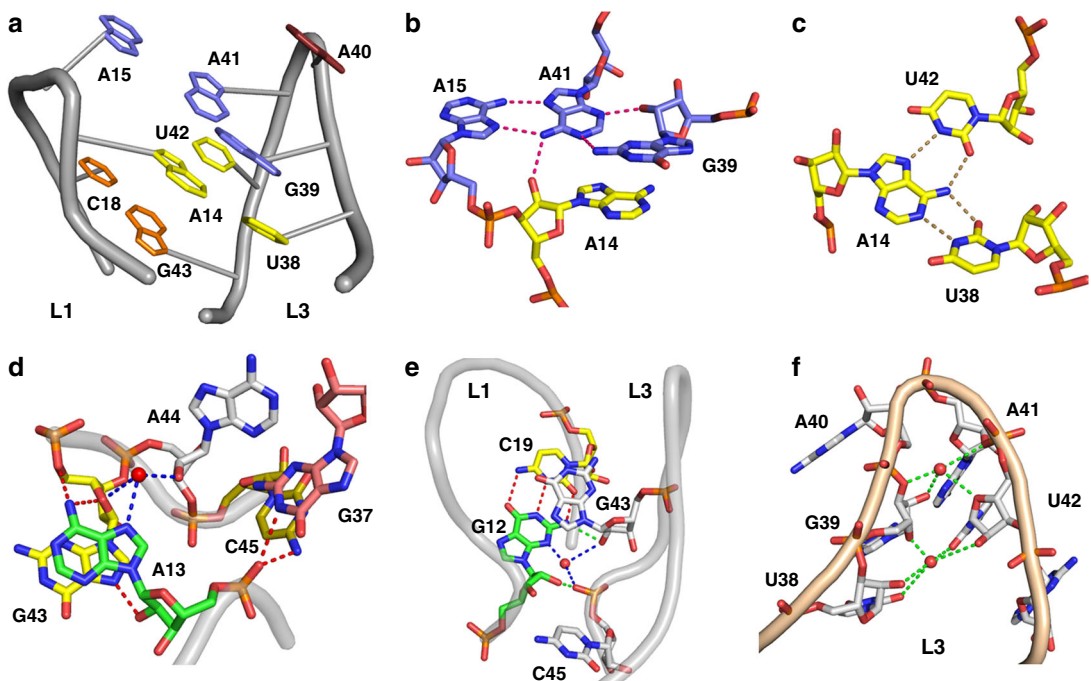

**Fig. 2** Loop–loop interactions. **a** Cartoon representation of loop L1 and loop L3. For clarity, fluorophore OTB-SO₃ is omitted. **b** A41 forms the hub of the A15A41G39 base triple, ligand-stacking platform. Shown are hydrogen-bonding interactions between A41 and adjacent nucleotides. **c** Hydrogen-bonding interactions in the UAU base triple that supports the A15A41G39 stacking platform from below. **d**, **e** A13 and G12 (shown in green) interactions with neighboring nucleotides via stacking and hydrogen-bonding. Red spheres represent water molecules, and blue dashed lines denote inferred water-mediated hydrogen bonds. **f** Water-mediated hydrogen-bonding networks stabilize the loop L3 conformation, facilitating a sharp backbone bend at A40. Red spheres represent water molecules and dashed green lines denote hydrogen bonds

sugars of the G39, A41, and U42 form water-mediated hydrogen bonds that presumably support the bending of the phosphate backbone to make the sharp U-turn at A40 and thereby place the G39 and A41 nucleobase planes parallel to each other. Sugar edges of the nucleotides U38 and U42 also interact with each other by forming water-mediated hydrogen bonds (Fig. 2f).

**Structure of the fluorophore-binding site**. The fluorophore-binding site resides at the top of the kissing loops L1 and L3 at the platform created by the A15-A41-G39 base triple (Fig. 3a, b and 2b) and further stabilized by the A14-U38-U42 base triple (Fig. 2c) underneath. The OTB-SO₃ fluorophore sits in a planar conformation predominantly upon the G39 and A41 nucleobases, which form stacking interactions with the benzothiazolium and benzoxazole rings, respectively (Supplementary Fig. 5). The orientation and conformations of the two rings in the OTB-SO₃ was ascertained by anomalous data collected using the single frequency for the sulfur, which confirmed higher occupancy for the sulfur in the benzothiazole ring (Supplementary Fig. 6). A sharp U-turn in L3 at A40 positions this nucleotide at the apex of the RNA fold, allowing the adenine ring to stack upon the benzothiazolium ring to complete a stacking "sandwich". Although both the thiazolium and oxazolium rings contain heteroatoms that could, in principle, accept hydrogen bonds, the fluorophore rings associate with the RNA solely through stacking interactions. Nevertheless, the sulfonate-bearing propyl side chain extends toward the stacking platform beneath to form bifurcated hydrogen bonds with the amidinium moiety of G39.

In the crystal lattice, two RNA aptamers from neighboring symmetric units pack against each other in a head-to-head fashion to bury 420 Å² of surface area (Fig. 3c). In this homodimer interface, the faces of the apex nucleobase A40, the benzoxazole ring, and the platform nucleobase A15 resemble

descending stairs that stack upon A15, the benzoxazole ring, and A40, respectively, of the neighboring aptamer to create a four-layer stacking wall (Fig. 3d). The nucleobases of the base-triple platform A15-G39-A41 occupy the first and fourth layers, whereas the middle two layers each consist of the fluorophore rings nestled against A40 from the neighboring monomer, resulting in burial of nearly all (92%) of the OTB-SO₃ surface area. Hydrogen bonds between the A40 2′OH and A15 N1 (Supplementary Fig. 7) further support this lattice interaction allowing the benzoxazole rings of the respective fluorophores to stack on each other (buried surface area (BSA): 104 Å², contributing 25% of the total interface BSA).

The recent crystal structure and associated solution studies of the Corn RNA aptamer revealed a preformed homodimer ($K_d < 1$ nM) in which the interface between monomers binds and activates fluorescence of DFHO[36]. Our crystal structure raised the possibility that DIR2s could function analogously. To address this possibility, we investigated whether the DIR2s aptamer forms a homodimer in solution using several experimental approaches. First, we titrated a fixed concentration of fluorophore (10 μM) with increasing concentrations of the aptamer and observed a linear increase in fluorescence ($R^2 = 0.996$) over the 0–2.5 μM concentration range (Fig. 4a, insert). A requirement for dimerization would be expected to yield a parabolic rather than linear dependence on aptamer concentration, unless the aptamer dimerizes at low nanomolar concentrations. Fluorescence reached a plateau as the aptamer reached a concentration equivalent to that of the dye present, ~ 10 μM (Fig. 4a and Supplementary Fig. 8), consistent with formation of a one to one (or 2:2) DIR2s: OTB–SO₃ complex in agreement with continuous variation experiments reported previously[40]. Second, we performed solution small-angle X-ray scattering (SAXS) measurements at various DIR2s–OTB–SO₃–Fab ternary complex concentrations.

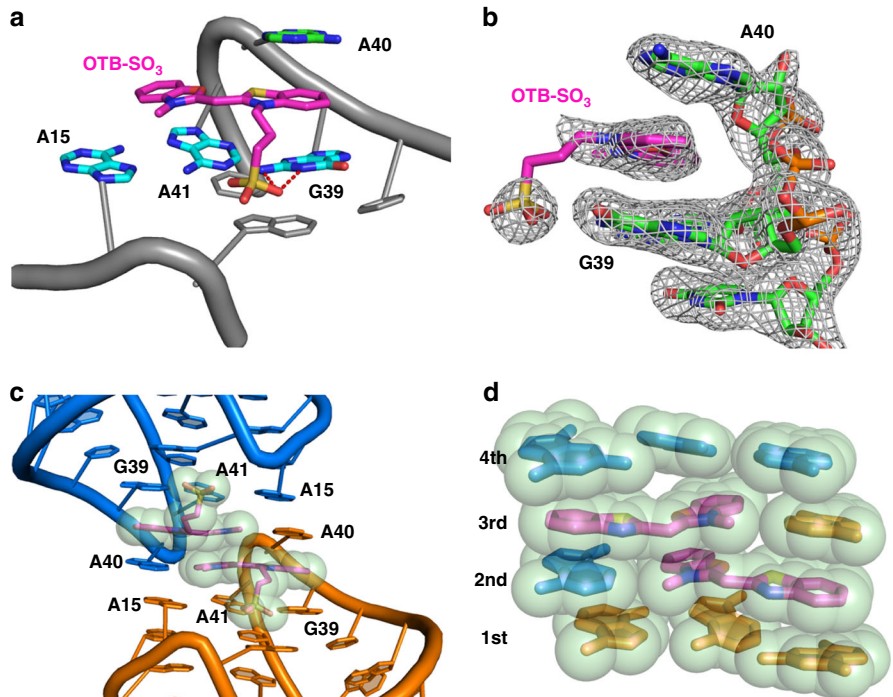

**Fig. 3** Structural basis of OTB-SO$_3$ fluorophore binding to DIR2s RNA aptamer. **a** Cartoon representation of the ligand-binding site showing OTB-SO$_3$ ligand stacked upon G39 and A41 and sandwiched by A40 from the top. The propylsulfonate side chain interacts with G39 through hydrogen bonds. **b** Side view showing a claw-like structure of the ligand-binding site represented by sticks superimposed on 2|$F_o$| − |$F_c$| electron density map. **c** RNA dimerization by head-to-head stacking with the neighboring asymmetric unit (shown in blue) observed in the crystal packing. At the interface, OTB-SO$_3$ fluorophores from two molecules (shown by sticks and transparent spheres) stack upon each other, whereas A15 and A40 form reciprocal stacking interactions with the other protomer. **d** A four-layer stacking interaction at the interface of the RNA dimer. Only nucleobase and fluorophore rings are shown; the propylsulfonate substituents and other RNA structural components are omitted for clarity

Fitting the observed scattering profiles against calculated scattering from the crystal structure indicates that, at increasing concentrations, the RNA exists as a monomer:dimer mixture (80:20% at 4 mg/mL). However, at the lower concentration (1 mg/mL, ≈ 52 μM), which is >10- to 15-fold higher than the RNA concentrations used in fluorescence studies performed in vitro and in vivo, the RNA populates predominantly (> 90%) the monomeric form. (Fig. 4b and Supplementary Fig. 9 and Supplementary Table 1). Third, we performed size exclusion chromatography (SEC) of DIR2s:OTB–SO$_3$ complex over the concentration range of 1–20 μM and compared its elution volume with 21 kDa and 39 kDa molecular weight (MW) RNA standards. As expected for the monomeric form of the aptamer (MW ~ 18 kDa), we observed a single peak that eluted slightly later than the 21 kDa standard (Supplementary Fig. 10). These biophysical analysis experiments demonstrate that at concentrations used in fluorescence measurement experiments ( < 5 μM) the DIR2s aptamer exists predominantly as a monomer that binds OTB-SO$_3$ dye in 1:1 stoichiometry in solution.

As our crystallization constructs bind and activate fluorescence of both DIR and OTB dyes, we wanted to test whether these fluorophores bind competitively. We performed a binding competition between OTB-SO$_3$ and DIR dyes by titrating the DIR dye with a fixed concentration of the DIR2s aptamer–OTB–SO$_3$ complex and monitored fluorescence at two different wavelengths. We observed a decrease in fluorescence of the OTB-SO$_3$ dye as a function of increase in concentration of DIR dye (Fig. 4c), demonstrating that the fluorophores bind to the DIR2s aptamer in a mutually exclusive manner. This and several other independent observations are consistent with both dyes binding to the same site within the aptamer (see Discussion). Nevertheless, as we do not have the structure of the DIR-SO$_3$ dye

bound to the aptamer, we cannot formally exclude the possibility that the dyes bind to distinct sites and allosterically impart mutual exclusivity. As a further assessment of ligand-binding promiscuity, we tested whether DIR2s could activate fluorescence of TO1-Biotin, the cognate fluorophore for the RNA Mango aptamer[19]. Under identical conditions, DIR2s activated TO1-Biotin fluorescence to nearly the same intensity as did RNA Mango (Supplementary Fig. 11), showing that DIR2s can bind and activate other structurally similar fluorophores.

In the monomeric form of the complex, the benzoxazole ring of the fluorophore lacks the spatial restriction and upper face stacking interaction imposed by A40 from the opposite aptamer, raising the question of whether the structure accurately reflects functional features of the aptamer. To test this, we created DIR2s variants with perturbed ligand-binding sites and measured their ability to activate OTB-SO$_3$ fluorescence (Fig. 4d and Supplementary Fig. 12). Initially, we prepared single-residue mutations such as A40 to U and A41 to U, however, neither mutation affected the fluorescence activation compared with the WT aptamer. Possibly in the mutant A40U, U can stack on the fluorophore from the top. Previous studies have noted that U/T can provide stacking interactions to stabilize aromatic ligands[48]. The aptamer bearing the A41U mutation might maintain a stacking platform for the ligand binding by forming an U41-A15 base pair that can still interact with G39. Tolerance of these mutations lead us to test multiple-residue mutations. As expected, simultaneous mutation of the three purine nucleotides that form the stacking platform A40-A41-G39 to uridine residues completely abolished fluorescence activation. Simultaneous mutation of A15 to C and A40 and A41 to G also completely abolished fluorescence. To probe the contribution of the G39 and A40 nucleobases individually, we constructed aptamer variants

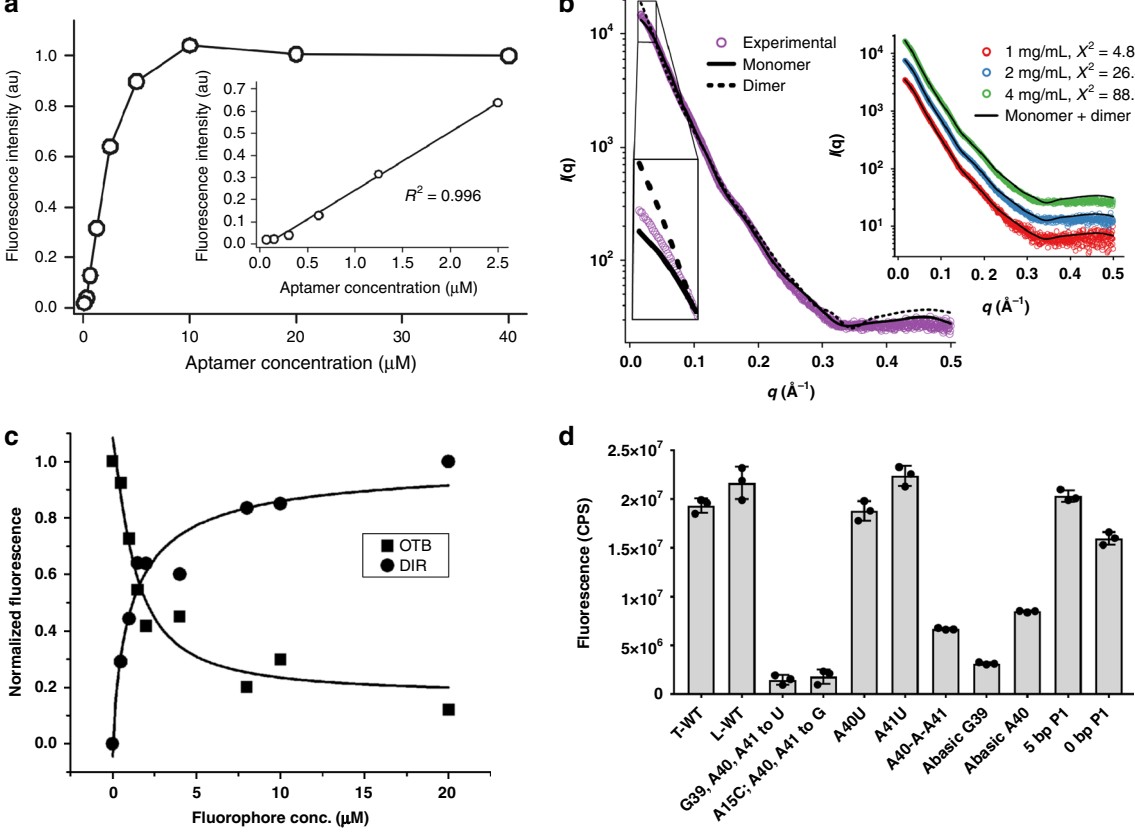

**Fig. 4** Functional analysis of the DIR2s aptamer. **a** Fluorescence activation of OTB-SO$_3$ dye as a function of DIR2s RNA aptamer concentration. Insert shows fluorescence signal increases linearly ($R^2 = 0.996$) with aptamer concentration in the range of 0–2.5 µM indicating, 1:1 RNA:fluorophore complex formation in solution. The concentration of OTB-SO$_3$ was 10 µM. **b** Solution phase analysis of the aptamer with SAXS. Fitting of experimental SAXS profile (scattering vector, $q = 4\pi \sin \theta/\lambda$) obtained by merging the datasets from 1 mg/mL, 2 mg/mL, and 4 mg/mL concentrations for RNA•OTB-SO$_3$•Fab complex to a monomer ($\chi^2 = 30.1$) or a dimer ($\chi^2 = 161.4$) model from the crystal structure. Insert shows improved fitting of the experimental datasets for each concentration when a fitting that accounts for both monomeric and dimeric forms of the RNA in the solution. Details of SAXS analysis for both RNA•OTB-SO$_3$•Fab BL3-6 and RNA•OTB-SO$_3$ complexes are provided in Supplementary Fig. 9 and Supplementary Table 1. **c** DIR-SO$_3$ and OTB-SO$_3$ dyes compete for binding and fluorescence activation by the DIR2s aptamer. DIR2s•OTB-SO$_3$ complex (500 nM RNA, 600 nM OTB-SO$_3$) was titrated with DIR-SO$_3$ at the indicated concentrations. **d** Importance of binding site nucleobases and the P1 stem for OTB-SO$_3$ fluorescence activation (RNA concentration 3 µM and OTB-SO$_3$ 6 µM) T and L denote transcribed and ligated aptamers, respectively. The error bars indicate mean and standard deviations from three consecutive measurements

containing an abasic ribose (ab) residue (which replaces the entire nucleobase with a hydrogen atom) by enzymatic ligation of synthetic RNA oligonucleotides. Both G39ab and A40ab variants exhibited diminished fluorescence signal compared with the WT aptamer. From the structure, we hypothesized that insertion of an additional purine nucleotide between A40 and A41 might be able to provide stacking interaction to the benzoxazole ring of the OTB from the top; however, this insertion had an adverse effect on OTB-SO$_3$ fluorescence. Together these results support the functional significance of these nucleotides and suggest that that the crystal structure captures the aptamer–OTB–SO$_3$ complex in a functionally relevant conformation. To test the functional significance of the P1 stem, we constructed truncates of the DIR2s aptamer lacking 5 base pairs (nucleotides 1–5 and 56–60) and 10 base pairs from the end (nucleotides 1–10 and 51–60). For the latter construct, we replaced U11 and U20 with G and C, respectively. The construct with the five base pair P1 stem retained fluorescence similar to that of WT aptamer, and the construct lacking P1 entirely showed slightly lower fluorescence (ca. 80%), indicating that the P1 stem has little functional importance (Fig. 4d and Supplementary Fig. 13). Perhaps the P1 stem emerged from the selection to sequester the

corresponding nucleotides and prevent them from interfering with the L1–L3 loop interaction.

**The crystal structure of apo form of the aptamer.** We also determined the crystal structure of the aptamer construct complexed with Fab BL3-6 in absence of fluorophore at 2 Å resolution (Table 1). We solved the structure using molecular replacement with Fab BL3-6 coordinates (PDB: 3IVK) as a search model. The apo form of the DIR2s aptamer adopts a highly similar architecture to its ligand-bound form (Fig. 5a and Supplementary Fig. 14), with the two loops L1 and L3 held together via the same hydrogen-bonding and stacking interactions. The two structures superimpose well (r.m.s. deviation 1.816 Å) with the exception of the aforementioned nucleotides at the end of the helix P1 that could not be modeled accurately and local changes in the ligand-binding site. The most prominent difference occurs at the apex nucleotide A40; relative to its position in the ligand-bound form, A40 rotates by 180° around the glycosidic bond in apo form and points upward in ca. 50° angle relative to the A40 plane of the ligand-bound form (Fig. 5b, c), making a hydrogen bond with the aptamer in the neighboring asymmetric unit. Nucleotide G39 in the apo form lays offset by 20° to that of the G39 plane in the

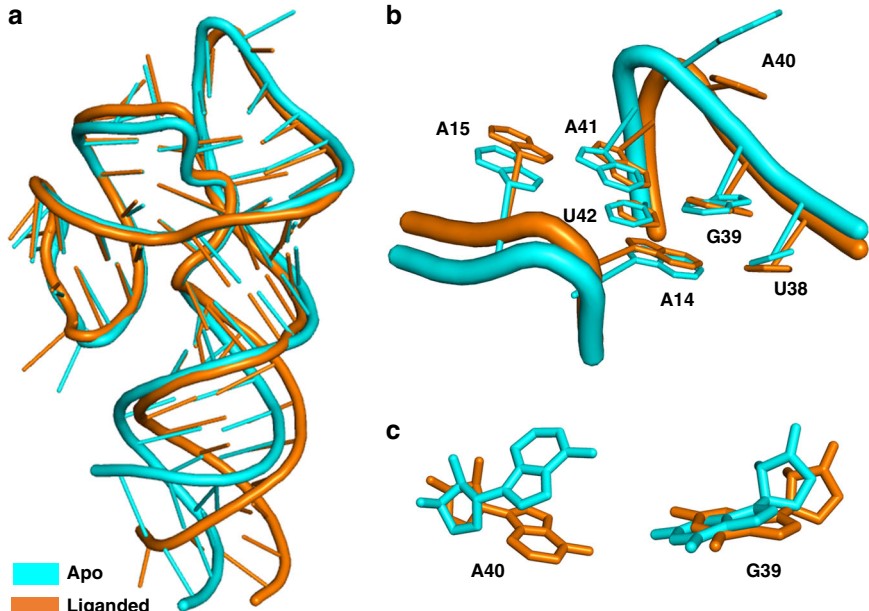

**Fig. 5** OTB-SO$_3$ fluorophore causes only local structural changes in the DIR2s aptamer. **a** Overlay of the DIR2s aptamer structures obtained in the presence (brown) and absence (cyan) of the OTB-SO$_3$ fluorophore (r.m.s. deviation = 1.816 Å on RNA). **b** Overlay of the binding site in the presence (brown) and absence (cyan) of the OTB-SO$_3$ fluorophore. **c** Zoomed views of A40 and G39 in apo (cyan) and fluorophore-bound form (brown). In the apo form, A40 is flipped 180° relative to the A40 in the fluorophore-bound form. PDB ID: 6DB9

OTB-bound form (Fig. 5c). Both loops L1 and L3 of the apo form are shifted toward stem P3 by 2.5 Å compared with the OTB-bound form. In the crystal lattice of the apo form, aptamers are arranged in head-to-tail stacking, in contrast to the OTB-bound form, where alternating head-to-head and head-to-tail interactions occur, possibly reflecting the additional stacking interactions provided by bound ligand.

## Discussion
Using Fab BL3-6 and its cognate RNA hairpin motif as a crystallization module[44], we crystallized and solved the crystal structures of both apo and ligand-bound forms of the DIR2s aptamer. In neither case did we observe crystals in the absence of Fab BL3-6, suggesting that the Fab facilitates crystal lattice formation. In the crystal lattice, the Fab mediates a major portion of the crystal contacts (87%), whereas RNA contributed only 13% through RNA–RNA interactions. In addition, the Fab facilitated structure determination by allowing the use of molecular replacement to solve the phasing problem.

The co-crystal structure of the DIR2s aptamer and supporting biochemical studies reveal a compact, tuning fork-like architecture comprised of a helix and two stem-loops oriented parallel to each other. Interactions between the two loops create the fluorophore-binding site, which consists of an unpaired nucleotide (A40) that caps the benzothiazolium ring from the top and a base triple formed by three purines (A15, G39, and A41) that provide a stacking platform for the two rings of OTB to sit in a planar configuration. Together the platform and the unpaired nucleotide form a claw-like structure that clamps down on the ligand. OTB's propylsulfonate projects downward from the open end of the claw to position the sulfonate moiety for chelation by the G39's amidinium group. In the crystal lattice, two complexes homodimerize via head-to-head stacking interactions involving the ligand-binding sites, resulting in nearly complete surface burial of the fluorophore. Dimerization governed by stacking interactions resembles a feature of the recently reported Corn RNA aptamer[36], which binds its DFHO ligand with 2:1

aptamer:fluorophore stoichiometry. However, Corn functions as a dimer in solution, whereas DIR2s, which buries only half as much surface area at the dimer interface (880 Å$^2$ vs. 420 Å$^2$ for DIR2s with OTB dye), remains monomeric at micromolar concentrations.

According to the structure, the monomeric form leaves the top face of the benzoxazole ring uncapped and exposed to solvent. This open feature likely relaxes the ligand specificity and accounts for the ability of the aptamer to bind both OTB-SO$_3$, DIR-SO$_3$, and TO1-Biotin. If the aptamer functioned as a dimer, the snug fit for OTB-SO$_3$ at the dimer interface in the crystal would be expected to preclude binding and fluorescence activation of DIR, the actual ligand targeted in the selection. Indeed, crystallization trials in the presence of DIR-SO$_3$ did not yield crystals of the ligand-bound form. DIR-SO$_3$ and OTB-SO$_3$ bind the DIR2s aptamer competitively and exhibit similar responses to mutation (Fig. 4c, d and Supplementary Fig. 12), consistent with the possibility that they occupy the same binding site. Although we cannot rule out the possibility that the competition arises allosterically through binding at distinct sites, both ligands contain a positively charged ring and propylsulfonate substituent (Fig. 1a) that could occupy analogous positions within the ligand-binding site. In the case of DIR, this would leave the dimethylindole ring to extend over the stacking platform with one face exposed. Just as the geminal dimethyl group of DIR prohibits nonspecific intercalation into nucleic acid duplexes, its presence during selection may have disfavored the emergence of a binding site that caps the indole ring from the top. The lack of a complete stacking surface above OTB-SO$_3$ might also account for its somewhat lower fluorescence quantum yield when bound to the aptamer ($\phi_f = 0.51$) compared with when bound to certain scFv proteins selected from a yeast-displayed library for binding to this particular fluorogen ($\phi_f = 0.84$–$1.00$)[49].

The absence of a G-quadruplex in the DIR2s fluorophore-binding site is unique among fluoromodules as RNA mimics of green and red fluorescent protein Spinach[34,35] and Corn[36], respectively, as well as RNA Mango[37], all contain a G-quadruplex in their fluorophore-binding site. The MG aptamer lacks a G-

quadruplex motif, but the fluorophore-binding site contains a non-G-quartet base quadruple[50,51]. In contrast, the fluorophore-binding site of DIR2s aptamer uses a base triple as a stacking platform and contains no tetrad. All fluoromodule aptamers identified thus far use nucleobase stacking and shape complementarity to bind their profluorescent ligands and restrict motion, which decreases nonradiative decay and thereby increases fluorescence quantum yield. Similar to previous observations for the Spinach aptamer[34], the DIR2s aptamer retains its global architecture in the absence of ligand and undergoes only a local structure perturbation to accommodate the fluorophore. Ligand binding to this largely preorganized aptamer requires a simple rotation about the glycosidic bond of A40, and thus, likely imparts only a modest entropic penalty.

In summary, the DIR2s aptamer recognizes planar, aromatic chromophores without the use of a G-quadruplex or nucleobase tetrad, thereby expanding the scope of fluorogenic RNA activation. Our structures streamline RNA construct and ligand design for future applications of this fluoromodule and provide a physical framework to guide efforts to engineer DIR2s aptamers and ligands with improved photophysical and recognition properties.

## Methods

**RNA synthesis and purification**. The double-stranded DNA templates for transcription reactions were prepared by PCR amplification of the single-stranded DNA oligomer purchased from Integrated DNA Technologies (IDT). The first two nucleotides of the reverse primer contained 2′-OMe modifications to reduce transcriptional heterogeneity at the 3′ end[52] (Supplementary Table 2). RNA was prepared by in vitro transcription for 3 h at 37 °C in buffer containing 40 mM Tris-HCl pH 7.9, 2 mM spermidine, 10 mM NaCl, 25 mM MgCl₂, 10 mM DTT, 30 U/mL RNase Inhibitor, 2.5 U/mL TIPPase, 4 mM of each nucleoside triphosphate (NTP), DNA template 30 pmol/mL, 40 µg/mL homemade T7 RNA polymerase. Transcription reactions were quenched by adding 10 U/mL RNase free DNase I (Promega, www.promega.com) and incubating at 37 °C for 30 min. After the phenol/chloroform/isopropanol, pH 4.3 extraction, the RNA was purified by denaturing polyacrylamide gel electrophoresis (dPAGE). Corresponding RNA band was visualized UV shadowing and excised from the gel. RNA was eluted overnight at 4 °C in 10 mM Tris, pH 8.0, 2 mM EDTA, 300 mM NaCl buffer. The buffer in eluted RNA was exchanged three times by pure water using 10 kDa cutoff size exclusion column (Amicon). RNA was collected, aliquoted into small fractions, and stored at −80 °C until further use.

DIR2s aptamers containing abasic site were prepared by chemical ligation of two short oligonucleotides "Donor" and "Acceptor" synthesized in house by solid-phase synthesis on a 1-µmol scale using an Expedite Nucleic Acid Synthesis System (8900) by following standard RNA synthesis protocols. The oligonucleotides were released from solid support with 3:1 NH₄OH/EtOH at 55 °C for 8 h, desilylated with 300 µL 6:3:4 N-methylpyrrolidinone/triethylamine/triethylamine-3HF at 65 °C for 2 h and precipitated by n-BuOH. The oligomers were further purified by dPAGE, collected in pure water and stored at −80 °C until further use.

**Enzymatic ligation**. The Donor oligonucleotide was 5′-end phosphorylated with T4 polynucleotide kinase (New England Biolabs, NEB) in the presence of excess ATP at 37 °C for about 1 h followed by heat inactivation of the enzyme at 60 °C for 20 min. For the ligation reaction, the DNA splint oligonucleotide was mixed with donor and acceptor RNA oligonucleotides at a 1:1.5:1.1 ratio of donor/acceptor/splint and annealed in the presence of monovalent ions by heating to 90 °C and gradual cooling to 20 °C over a period of 1 h. After annealing, T4 RNA ligase buffer (50 mM Tris-HCl, 2 mM MgCl₂, 1 mM DTT, 400 µM ATP, pH 7.5) and T4 RNA ligase 2 (NEB) were added, and the resulting reaction mixture was incubated at room temperature for 12 h. Ligation reaction mixtures were purified by 15% dPAGE as described above and concentrations for pure oligos were determined using Nanodrop (Thermo Scientific).

**Fluorescence determination**. The fluorescence emission of DIR2s or mutant aptamers in complex with either DIR or OTB–SO₃ dyes (excitation and emission wavelengths for OTB-SO₃ 380 nm and 426 nm, and DIR 600 nm and 650 nm, respectively) were measured at 20 °C, using a Fluorolog-3 spectrofluorometer equipped with a thermo controller (Horiba Inc.) and reported results were the average of three measurements. The samples for fluorescence measurements were prepared by following a standard refolding protocol: the RNA was denatured by heat treatment for 1 min at 90 °C in deionized water followed by refolding in the buffer containing 50 mM Tris pH 7.4, 150 mM NaCl, 5 mM MgCl₂ (supplied as 10×) for 15 min at 50 °C, 10 min at 20 °C and incubated in the presence of dye for 30 min at 20 °C.

**Crystallization of Fab–RNA complex**. An aliquot of RNA sample was refolded in a buffer containing 50 mM Tris pH 7.4, 150 mM NaCl, 5 mM MgCl₂ (supplied as 10×). For refolding, RNA was heated at 90 °C for 1 min in deionized water and then incubated at 50 °C for 15 min in folding buffer followed by incubation at room temperature for 10 min. The refolded RNA was then incubated in the presence of dye for 30 min at 20 °C followed by 1.1 equivalents of the BL3-6 Fab (expressed as soluble protein in phagemid as an expression vector and, purified by affinity and ion exchange chromatography using protein A, G, and Heparin columns (GE Healthcare), respectively[43] at room temperature for 30 min and concentrated to 6 mg/mL using 10 kDa cutoff, Amicon Ultra-15 column. The formation of Fab–RNA complex was confirmed by native polyacrylamide gel electrophoresis. To decrease the number of nucleation events, RNA was then passed through the 0.2 µm cutoff, Millipore centrifugal filter units. A Mosquito liquid handling robot (TTP Labtech) was used to set up high-throughput hanging drop vapor diffusion crystallization screens at room temperature using commercially available screening kits from Hampton Research, Sigma, and Jena Bioscience. The best-diffracting crystals of the OTB-SO₃–DIR2s–Fab complex were obtained in a condition from the Natrix screen: 50 mM MES, pH 5.6, 10 mM MgCl₂, 1.8 M Li₂SO₄. The complex without the OTB-SO₃ fluorophore, the optimal condition was from the Jena Bioscience screen: 15 % v/v 2-Propanol, 50 mM MES pH 6.0 20 mM MgCl₂. Crystals appeared and grew to full size within 2 weeks in 1 µl + 1 µl hanging drops on siliconized glass slides. The crystals were socked in buffer containing 30% glycerol and saturated OTB dye for crystal with OTB dye before being flash-frozen in liquid nitrogen for data collection at Argonne National Laboratory.

**Structural data processing and analysis**. The X-ray diffraction datasets were collected at 100 K at the Advanced Photon Source NE-CAT section beamline 24-ID-C. All the datasets were then integrated and scaled using its on-site RAPD automated programs (https://rapd.nec.aps.anl.gov/rapd/). The structures were solved by molecular replacement. Initial electron density maps were obtained from Phaser[53] after searching for N-terminal variable domain and C-terminal constant domain of Fab BL3-6 (PDB: 3IVK) sequentially. DIR2s RNA and OTB-SO₃ ligand were built into the initial electron density maps using COOT[54]. The structures were refined using PHENIX[55] package. The restraint for OTB ligand was generated by using PRODRG[56]. Solvent-accessible surface area and area of interaction were calculated using PDBePISA[57]. All figures were made in PyMOL (Schrodinger).

**Fab–RNA binding affinity measurement by filter-binding assay**. Approximately 50,000 c.p.m. of 5′-³²P-radiolabeled DIR2s RNA was refolded in a buffer (900 µL) containing 50 mM Tris pH 7.4, 150 mM NaCl, 5 mM MgCl₂ at 50 °C for 10 min, 10 min at RT and added 1 U/µL RNase inhibitor (Amersham). The refolded RNA was incubated for 30 min with Fab BL3-6 ranging from 2 µM to 2 nM in a final volume of 40 µL. A nitrocellulose membrane (Whatman) at top and HyBond filter (Amersham) at bottom were placed in a 96-well Dot-Blot apparatus (Bio-Rad), and the wells were pre-equilibrated with 100 µL refolding buffer. The Fab–RNA mixture was applied, and the filter was washed with another 100 µL of refolding buffer. The membrane and filter were air-dried, exposed to Phosphorimager screens and scanned with a Typhoon Trio imager (GE Healthcare). The amount of radiolabeled RNA bound to each filter was quantified with ImageQuant software (Molecular Dynamics). Binding constants were obtained by fitting the fraction of nitrocellulose-bound RNA to the following Hill equation: $F = F_0 + F_{max} \left( \frac{[Fab]}{K_d + [Fab]} \right)^n$, where $K_d$ is the dissociation constant; $F_0$ and $F_{max}$ are the minimum and maximum fractions of RNA bound; and $n$ is the Hill coefficient.

**SAXS data collection and processing**. SAXS experiments were conducted on the SIBYLS beamline at the Advanced Light Source, Lawrence Berkeley National Laboratory, essentially as described previously[58,59]. RNA and Fab–RNA complex samples bound to the OTB dye were prepared and purified as described above. Sample preparation and measurements were performed in a buffer consisting of 50 mM Tris pH 7.4, 150 mM NaCl, and 5 mM MgCl₂. For each complex, data on three concentration samples, 1.0 mg/mL, 2.0 mg/mL, and 4.0 mg/mL were collected, along with buffer blanks before and after each sample. Data were collected as 32 0.3-s exposures, in the range $q = 0.01087$–$0.565291$. For buffer subtraction, averages of the 32 exposures of pre- and post-sample buffer were subtracted from each exposure of the sample. Each buffer-subtracted exposure was checked for radiation damage by examining the Gunier region using AUTORG[60], outlier exposures were removed, and the remaining exposures averaged using DAT-MERGE[61] to produce a final curve for each concentration/sample. Fitting of the Gunier region to calculate $R_g$ for each sample was performed using AUTORG[60]. For calculation of scattering from the crystal structure atomic coordinates and subsequent fitting to experimental data, ALMERGE[58] was used to merge the low- and high-q segments of the 1 mg/mL and 4 mg/mL samples, respectively, and this was used along with atomic coordinates as input to FOXS[62]. To improve the fit to the data, OLIGOMER[63] was used to fit a mixture of crystallographic monomer and dimer to each concentration, using the fits from FOXS[62] to monomer and dimer alone as the input form factors. A summary of data

collection parameters, programs used for data analysis, and fits to the data is presented in Supplementary Table 1.

## Data availability

Data supporting the findings of this manuscript are available from the corresponding author upon reasonable request. Atomic coordinates and structure files for the Fab-DIR2s aptamer–OTB–SO$_3$ complex and its apo form crystal structure have been deposited in the Protein Data Bank (http://www.pdb.org/) under accession code 6DB8 and 6DB9, respectively.

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

## Acknowledgements

We thank staff of the Advanced Photon Source at Argonne National Laboratory for providing technical advice on X-ray data collection and the staff of SIBYLS beamline at Lawrence Berkeley National Laboratory for SAXS data collection. We thank Prof. Phoebe A. Rice for useful suggestions during structure solving and members of the Piccirilli group for comments on the manuscript. N.-S. Li for synthesis of RNA oligonucleotides containing an abasic site and TO1-Biotin. This work was supported by grants from the US National Institutes of Health (R01AI081987, R01GM102489), the Chicago Biomedical Consortium with support from the Searle Funds at The Chicago Community Trust and Defense Threat Reduction Agency (HDTRA1-13-1-0004) to J.A.P.; Predoctoral Training Program in Chemistry and Biology T32GM008720 to B.P.W. The X-ray crystallographic work is based on research conducted at the Advanced Photon Source on the Northeastern Collaborative Access Team beamline, 24-ID-C, which are supported by a grant from the National Institute of General Medical Sciences (P41 GM103403) from the National Institutes of Health and Advanced Light Source beamline SIBYLS, all supported by US Department of Energy (DOE). This research used resources of the Advanced Photon Source, a US DOE Office of Science User Facility operated for the DOE Office of Science by Argonne National Laboratory under contract DE-AC02-06CH11357 and Advanced Light Source, a US DOE Office of Science User Facility operated for the DOE Office of Science by Lawrence Berkeley National Laboratory under Integrated Diffraction Analysis (IDAT) grant contract DE-AC02-05CH11231.

## Author contributions

S.A.S. and J.A.P. designed the project; S.A.S. and A.L. set up high-throughput crystallization experiments and optimized crystallization conditions and screened crystals; S.A.S. collected the data; Y.S. phased and solved the structures; S.A.S. prepared Fab and conducted biochemical and biophysical assays; S.A.S. collected SAXS data, J.R.F., D.K., S.A.S. and J.A.P. analyzed the SAXS data; B.P.W. prepared abasic RNA constructs; X.T., T.P.C., A.S.W., M.P.B. and B.A.R. provided dyes; S.A.S and J.A.P. analyzed the overall data and wrote the manuscript; all the authors critically reviewed the manuscript.

## Additional information

**Competing interests:** The authors declare no competing interests.

