## [Peer Review File · Nature Communications]

Reviewers' Comments:

Reviewer #1:

Remarks to the Author:

Piccirilli and co-workers use a Fab fragment to facilitate crystallization and solving of the structure of a new fluorophore-binding aptamer using molecular replacement. The aptamer binds two relatively unrelated fluorophores, dimethylindol red (DIR) and oxazole thiazole blue (OTB). The so-called "DIR2s" RNA aptamer is a fluorescent turn-on RNA that specifically binds DIR-SO3 and OTB-SO3. The authors succeed to obtain crystals of both of apo- and ligand-bound aptamer, and solve their structures. The aptamer structure provides insight into binding site of OTB-SO3. Also the crystal structure revealed intra-loop interactions as well as inter-loop (L1-L3) interactions. The authors then perform functional analyses, and the results suggest that the aptamer exists predominantly as a monomer that binds the OTB-SO3 ligand in 1:1 stoichiometry in a solution. Unlike the previously identified crystal structures of Spinach, Corn, and RNA Mango, the DIR2s has no G-quadruplex and the novel mechanism for the fluorophore binding. Overall, the authors well-describe structural properties of the DIR2s aptamer which are novel and supported by the crystal structure and biophysical analyses. However, there are some concerns that the authors should clarify before publish in Nature Communications.

Major comments

1. The authors report a resolution of 1.8 Å for the structure. Is this for the overall complex including the Fab protein fragment? What is the resolution for the RNA portion alone? This should be reported as well in the paper.
2. The authors mention that the aptamer, which was selected for binding to DIR-SO3, is promiscuous and also binds the fairly different structured OTB-SO3. This can be understood in terms of the binding site in the crystal structure. This leads to a few questions: (1) Is this anticipated by lack of a counter selection in the study? (2) Is lack of specificity advantageous in any way? (3) This finding anticipates that many fluorescent moieties with pendant sulfites might fluoresce with this aptamer. I would like to see this tried by the authors—it is a simple experiment that would elevate the paper beyond a structure explaining a function by making it have predictive power.
3. The authors performed titration experiments where the DIR2s aptamer is revealed to bind OTB-SO3 dye with 1:1 stoichiometry in their test condition. Then, the authors carried out SAXS experiments where the DIR2s aptamer is suggested to be predominantly as a monomer. However, there is no direct observation. The authors should perform size exclusion chromatography or EMSA to discuss the aptamer really exists as a monomer structure in a solution.
4. The competition experiment in Fig 4c, doesn't really show that the two dyes bind the same site. Competition could be allosteric between two sites.
5. The authors designed DIR2s variant to look at important nucleotides for the fluorescent binding. The mutant aptamers of A40U-A41U-G39U, and A15C-A40G-A41G depresses the fluorescent activation, indicating these nucleotides significantly contribute on the fluorescent binding. However, which nucleotide is critical for the binding is still unclear. The reader thinks a single mutation is the best way to analyze it.
6. The most interesting part of this paper is that this is the first fluorophore-binding and -enhancing aptamer that works without a G-quartet. The authors should consider putting this into the title of their paper. It is profound given that eukaryotic cellular conditions unfold G-quartets, and it is a non-obvious and non-predictable results because G-quartets in aptamers are not always easily predicted.
7. The similarity of the ligand-bound and ligand-free structures suggests a role for minimal entropy loss in ligand binding, especially given the paucity of interactions with the ligand (e.g. no interactions with the benzoxazole ring). It was surprising that this feature wasn't discussed.

8. The authors obtain their crystals in 1.8 M lithium sulfate. (a) This is given as LiSO₄ in the Materials and Methods but should be Li₂SO₄. (b) 3.6 M Li⁺ would preclude G-quadruplex formation. Does the structure change when K⁺ or Na⁺, which favor G-quadruplex formation, are added? This could be assayed by comparing fluorescence in these different monovalent salts at more reasonable (100 mM) salt concentration.

Minor comments

- I found it confusing and somewhat off-putting to find “DIR2s” in the title and opening line of the Abstract without any decoding or explanation. Later we are provided that this is a second-generation binder for a fluorophore called DIR, although I still don't know where the 's' comes from. Please explain earlier.
- The description of the Fab interactions on p4/5 is fairly ordinary and could be moved to the SI.
- Chemical structures of DIR-SO₃ and OTB-SO₃ are dually shown in Fig. 1 and Supplementary Fig. 1. The authors should delete the structure on Supplementary Fig. 1 to prevent confusion.
- The figure call outs in the ms. goes straight from Fig1 to Fig 3.
- Fig. 4d, what does the error bars mean?
- Supplementary Fig. 4 and Supplementary Fig. 9 are not used in the main text.
- Supplementary Fig. 9. Y axis is weird. Why do they need to change the scale change between 1×10^7 and 0? In other words, just report as 0.5×10^7 . Also Fig. 4d shows that the fluorescence for 0 bp P1-aptamer is ~20% smaller than that for 5 bp P1-aptamer. Despite the same condition, why the fluorescence for these aptamers has similar values in Supplementary Fig. 9?
- Fig 1b. Capitalization of the dyes is uneven.
- At one point the authors refer to the G as having an amidinium group and another as a guanidinium group. Either can be used, although I prefer the former, but the authors should be consistent.

Reviewer #2:

Remarks to the Author:

The authors report crystal structures of an in vitro evolved RNA aptamer that binds two different fluorescent compounds, dimethylindole red and oxazole thiazole blue, as well as the structure of the RNA in the absence of either ligand. The experimental crystallographic results all appear to be quite sound, and represented accurately. However, Figure 3b does not give us a good view of the quality of the electron density map and difference Fourier.

The three RNA crystal structures are quite similar to one another; the main difference in the apo structure is likely a minor artifact due to crystal contact interactions. Although several previous RNA aptamer structures that bind to fluorescent compounds have been previously determined, this RNA is unique in that it does not possess a G quadruplex platform upon which the fluorescent compound stacks. Instead, this RNA possesses a base triple (A15-A41-G39) as a stacking platform, which appears to be the primary claim to novelty of the paper, although the absence of the potential for formation of a G tetraplex was previously apparent from the sequence and secondary structure of the aptamer.

This claim bothers me a bit, because that triple is part of a coplanar array of four purines (A15-A41-G39-A14), which is clearly analogous to (but not identical to) a (purine) G-tetrad. In other words, it looks like the authors, in their zeal to make the structure sound unique, have overlooked or de-emphasized what is to me the most interesting result -- that there is more than one way to make a purine tetrad that supports fluorescent ligand binding. The paper would be much more interesting if this aspect was elaborated upon. What is doing on, electronically, with respect to the purine binding platform structures. Presumably, it has to do with pi-orbital conjugation. Perhaps it is beyond the scope of the current manuscript, but this should be elucidated, perhaps in collaboration with a computational quantum chemist.

Reviewer #3:

Remarks to the Author:

In this manuscript the authors solved the crystal structure of DIR2s aptamer in the apo and holo form (bound to the blue fluorescent oxazole thiazole blue, OTB-SO₃, fluorophore). The DIR2s aptamer has been previously shown to be a promiscuous RNA aptamer, binding to both blue fluorescent OTB-SO₃ and red fluorescent dimethylindole red (DIR-SO₃) and has been utilized to image epidermal growth factor receptor (EGFR) when fused to an EGFP-binding RNA aptamer. Even though the authors compared DIR2s to Spinach and Mango which have been demonstrated to be valuable tools for imaging RNAs in live cells, the usefulness of the DIR2s in RNA imaging has not been proven yet, which in my opinion reduces the significance of this work for a broader community. While the provided structure is certainly interesting, a structure of the red dye in complex with aptamer would be more relevant to potential users, as blue fluorescence is much more difficult to implement in biological systems. The stoichiometric mismatch between crystallographic (2:2) and solution experiments (1:1) concerns me. Specific comments are provided below:

1. It is not clear if replacing the UUCG tetraloop with GAAACAC sequence and binding of the Fab to the modified-aptamer has any effect on the binding affinity of the aptamer to the fluorophores. The authors only compared the fluorescence values, presumably in the presence of very high concentrations of the aptamer or dye (Figure S1c). K_d values should be reported here to really understand the influences of those changes.
2. In Figure 4a, why is there a data point with a negative fluorescence intensity?
3. In Figure 4a, authors show that when 10 μM of the fluorophore is titrated with up to 1.5 μM of DIR2s, they obtain a linear increase in fluorescence with slope of 1.08 and conclude that this shows 1:1 binding. First of all, the slope in this graph does not equal to 1. Fluorescence intensity has arbitrary units, and can be normalized to any value. This would change the calculated slope. Therefore, the value of the slope does not mean much. In addition, if the affinity for dimer formation (i.e., the 2:2 dye:aptamer complex) is very strong (i.e., K_d<2 nM), one would still observe a linear increase in this experiment. A more meaningful experiment is to do the same experiment with higher concentration of the DIR2s (for instance up to 30 μM). If you reach the plateau when 10 μM of DIR2s added, it means 1:1 binding (or 2:2). Similarly, if you reach the plateau when 20 μM of DIR2s added, it means 1:2 binding, etc.
4. In Figure 4c, authors show that the fluorescence of the DIR2s-OTB-SO₃ complex decreases upon addition of DIR dye, and conclude that both dyes bind to the same site. However, it is also possible that DIR dye binds to somewhere else, influencing the binding pocket of OTB-SO₃ and causing OTB-SO₃ to dissociate.
5. Permeability of the dyes?
6. Mismatch between title and content – the promiscuity aspect is only speculated about, not experimentally demonstrated. To demonstrate the structural basis for promiscuous binding one would need more ligand-bound structures than 1.
7. If two dyes stack directly upon each other (as shown in Fig. 3c,d) this stacking will influence the photophysics. It might result in a quenched dimer or in an excimer, but it will most likely not have the same fluorescence properties as the monomeric aptamer-dye complex with the top face exposed to water. Recording fluorescence spectra of the crystals, and also of 1:1 mixtures of

aptamer and dye from very dilute to very concentrated should more convincingly resolve this dimer vs. monomer issue.

8. Citations of fluorogenic aptamers are incomplete.

9. Last sentence of abstract inaccurate. Correct statement would be "unique among fluorogenic RNAs with known 3D structure".

10. Minor textual inconsistencies (head-to-tail vs. end-to-end, p. 9)

Point-by-point response to reviewers

Reviewer #1 (Remarks to the Author):

Piccirilli and co-workers use a Fab fragment to facilitate crystallization and solving of the structure of a new fluorophore-binding aptamer using molecular replacement. The aptamer binds two relatively unrelated fluorophores, dimethylindole red (DIR) and oxazole thiazole blue (OTB). The so-called “DIR2s” RNA aptamer is a fluorescent turn-on RNA that specifically binds DIR-SO₃ and OTB-SO₃. The authors succeed to obtain crystals of both of apo- and ligand-bound aptamer, and solve their structures. The aptamer structure provides insight into binding site of OTB-SO₃. Also, the crystal structure revealed intra-loop interactions as well as inter-loop (L1-L3) interactions. The authors then perform functional analyses, and the results suggest that the aptamer exists predominantly as a monomer that binds the OTB-SO₃ ligand in 1:1 stoichiometry in a solution. Unlike the previously identified crystal structures of Spinach, Corn, and RNA Mango, the DIR2s has no G-quadruplex and the novel mechanism for the fluorophore binding. Overall, the authors well-describe structural properties of the DIR2s aptamer which are novel and supported by the crystal structure and biophysical analyses. However, there are some concerns that the authors should clarify before publish in Nature Communications.

We are thankful to the reviewer for the positive comments and valuable insights and suggestions. Responses to the specific comments and issues raised are provided in the following.

Major comments

1. The authors report a resolution of 1.8 Å for the structure. Is this for the overall complex including the Fab protein fragment? What is the resolution for the RNA portion alone? This should be reported as well in the paper.

Yes, this is the overall resolution including Fab protein and RNA. Each reflection is contributed to by all atoms in the crystal, therefore we cannot separate them and provide the resolution for RNA alone. However, we have added a figure to Supporting Information (Supplementary Figure 16) showing the B-factors corresponding to each region.

Figure: A cartoon representation of the structure Fab•DIR2s•OTB-SO₃ complex colored according to B-factors of each region. OTB-SO₃ dye has been omitted for clarity.

2. The authors mention that the aptamer, which was selected for binding to DIR-SO₃, is promiscuous and also binds the fairly different structured OTB-SO₃. This can be understood in terms of the binding site in the crystal structure. This leads to a few questions:

(1) Is this anticipated by lack of a counter selection in the study?

Originally, the DIR2s aptamer was selected against DIR-SO₃ dye using a DIR-SO₃ immobilized column for selection as reported in *J. Am. Chem. Soc.* 139, 9001-9009 (2017) by the Armitage group. A negative selection was done using a column lacking immobilized DIR-SO₃, however, no counter selection was done. Notably, a protein counterpart to the DIR RNA aptamer, an scFv protein selected previously by the same group for binding DIR dye, also showed promiscuous binding to other classes of cyanine dyes (*J. Am. Chem. Soc.* 2008, 130, 12620–12621). Based on these observations, it would seem that the promiscuous behavior is a possible outcome.

(2) Is lack of specificity advantageous in any way?

The promiscuity allows creation of a single RNA fusion where the dye-binding aptamer is tagged to something else but not be tied down to one color. For example, in the Armitage JACS paper, the aptamer was fused to an EGFR-binding aptamer and used to image EGFR on cell surfaces. This allows use of either the blue or red dye, flexibility that might be important if you are combining the aptamer with another color (e.g. a separate, FP-tagged protein such as RFP or another RNA fluoromodule). You can also add one dye at the start of an experiment, then switch over to the other dye at a later point. As demonstrated in the original report (*J. Am. Chem. Soc.* 2008, 130, 12620–12621), DIR2s aptamer in combination with DIR and OTB dyes has been used to follow cell surface expression and internalization of the endogenous receptor EGFR, distinguishing between two populations based on red (DIR) or blue (OTB) colored dye present at different time points. In this experiment, the aptamer was fused to another aptamer that binds to EGFR and triggers its endocytosis. First, cells were incubated for 30 min with the fused aptamer in the presence of OTB-SO₃ at 37 °C (a temperature permissive for endocytosis). Imaging the cells at this stage cannot easily distinguish whether the EGFR receptors remain on the surface or have been internalized. However, if the cells are washed to remove external fusion aptamer and dye and then stained with fusion aptamer bound to DIR dye on ice to prevent further internalization, a strong DIR dye signal is observed at the cell surface whereas the OTB-SO₃ signal is restricted to the internalized receptor pool. If only the original OTB-SO₃ could be used in the second incubation, it would be difficult to distinguish internalized receptors from non-internalized (cell surface) receptors. This experiment demonstrates that two populations of an endogenous biological target (i.e., EGFR) can be discriminated using a single affinity reagent that can be labeled separately with blue or red fluorogenic dyes.

Another advantage, as described recently for the SRB-2 fluoromodule (Murat Sunbul, Andres Jäschke; SRB-2: a promiscuous rainbow aptamer for live-cell RNA imaging, *Nucleic Acids Research*, gky543, <https://doi.org/10.1093/nar/gky543>), is that the promiscuity provides an opportunity to vary fluorophore structural elements systematically. The resulting structure-activity relationships can then be used to rationally design probes with improved properties for live-cell imaging.

(3) This finding anticipates that many fluorescent moieties with pendant sulfites might fluoresce with this aptamer. I would like to see this tried by the authors—it is a simple experiment that would elevate the paper beyond a structure explaining a function by making it have predictive power.

We thank the reviewer for the suggestion. We performed the binding experiment with a structurally similar fluorophore TO1-Biotin (chemical structure below), which is the cognate fluorophore for the RNA Mango aptamer. The DIR2s aptamer activates TO1-Biotin fluorescence to nearly the same intensity as does RNA Mango (see figure below), showing that DIR2s aptamer could bind other structurally similar fluorophores. Fluorophores containing pendent sulfites are not available commercially, and their synthesis requires substantial time and labor, making their investigation more appropriate for future study.

A sentence explaining this result and a figure (Supplementary Figure 11) have been added to the main text section, structure of fluorophore binding site page 8 and Supporting Information, respectively.

Figure: Chemical structure of TO1-Biotin.

Figure: Florescence enhancement of TO1-Biotin dye by its cognate (RNA Mango) and DIR2s RNA aptamers. The λ_{\max} for emission spectrum for TO1-Biotin only, TO1-Biotin with mango aptamer and TO1-Biotin with DIR2s aptamer were 535 nm, 544 nm and 539 nm, respectively.

3. The authors performed titration experiments where the DIR2s aptamer is revealed to bind OTB-SO3 dye with 1:1 stoichiometry in their test condition. Then, the authors carried out SAXS experiments where the DIR2s aptamer is suggested to be predominantly as a monomer. However, there is no direct observation. The authors should perform size exclusion chromatography or EMSA to discuss the aptamer really exists as a monomer structure in a solution.

We thank the reviewer for this important suggestion. We have performed size exclusion chromatography as requested by the reviewer. The DIR2s RNA aptamer (MW \approx 18 kDa) elutes from a Superdex 200 10/300 GL column (GE healthcare) later than reference RNA standards of 21 and 39 kDa, consistent with the monomeric state (see figure below). The elution time remains unchanged over the concentration range tested. Thus, we find no evidence of aggregation or dimerization with increasing concentration of the aptamer•dye complex up to 20 μ M. The figure showing this result has been added to the Supplementary Information as Supplementary Figure 10.

Figure: Size exclusion chromatography (SEC) of DIR2s aptamer at concentrations ranging from 1 to 20 μM in complex with 2 equivalents of OTB-SO₃ dye for each concentration. RNA standards with molecular weight (MW) ~21 kDa (Red) and ~39 kDa (green) serve as the references for the monomeric (~18 kDa) and dimeric (~36 kDa) forms of the aptamer, respectively. Over the concentration range tested, DIR2s RNA elutes as a single peak slightly slower than 21 kDa reference (see vertical dotted line for guidance) expected for the monomeric form of the aptamer in the solution. All SEC experiments were carried out in a 50 mM Tris pH 7.4, 150 mM NaCl, 5 mM MgCl₂ buffer at 4°C using a Superdex 200 10/300 GL column (GE healthcare).

4. The competition experiment in Fig 4c, doesn't really show that the two dyes bind the same site. Competition could be allosteric between two sites.

This is a valid point also raised by another reviewer (see also response to comment 4 from reviewer 3). As we do not have the structure of the DIR-SO₃ dye bound to the aptamer, the most we can say from a competition experiment is that the data are 'consistent' with binding to the same site. Several other independent observations are

also consistent with a single binding site. Comparison of the OTB-SO₃ bound form to the apo form indicates that the RNA is largely preorganized for OTB-SO₃ dye binding; e.g. there is no evidence of a conformation change at a site other than where the dye binds. The structure of the OTB-SO₃ bound form itself elicits the expectation that other dyes could bind in the same site, particularly in the region occupied by the benzoxazole ring of OTB-SO₃. Additionally, both DIR-SO₃ and OTB-SO₃ binding exhibited essentially the same response to mutations. Binding in separate sites would mean that the aptamer, which was selected for binding to DIR-SO₃, would contain a distinct site for OTB-SO₃. However, binding sites for these dyes not seem to occur ubiquitously in RNA as the discovery of this aptamer required several rounds (14) of in-vitro selection and enrichment. Although these independent observations support the idea that the binding sites for the two dyes overlap at least partially, we recognize that we still do not have a structure of DIR-SO₃ bound to the aptamer that would resolve this issue. We have noted this qualification in the revised manuscript and also modified the title.

Figure: Important nucleobases at the binding site and the P1 stem that contribute for DIR-SO₃ binding and fluorescence activation. RNA and DIR-SO₃ concentrations were 3 μ M and 6 μ M, respectively.

Figure: Important nucleobases at the binding site and the P1 stem that contribute for OTB-SO₃ binding and fluorescence activation. RNA and OTB-SO₃ concentrations were 3 μM and 6 μM, respectively.

5. The authors designed DIR2s variant to look at important nucleotides for the fluorescent binding. The mutant aptamers of A40U-A41U-G39U, and A15C-A40G-A41G depresses the fluorescent activation, indicating these nucleotides significantly contribute on the fluorescent binding. However, which nucleotide is critical for the binding is still unclear. The reader thinks a single mutation is the best way to analyze it.

We had prepared additional single residue mutations such as A40 to U and A41 to U and tested fluorescence activation of OTB-SO₃ dye with these mutants. However, neither mutation showed a defect relative to the WT aptamer (these results are shown in Figure 4d), leading us to test multiple-residue mutations. Possibly in the mutant A40U, U can stack on the fluorophore from the top. Previous studies have noted that U/T can provide stacking interactions to stabilize aromatic ligands (Nucleic Acids Research, 2012, Vol. 40, 3732–3740). The aptamer bearing the A41U mutation might maintain a stacking platform for the ligand binding by forming an U41-A15 base-pair that can still interact with G39. The tolerance of the aptamer to these point mutations led us to prepare more drastic point mutations using abasic ribose, which replaces the nucleobase with a hydrogen atom. We prepared the two single mutants containing an abasic site at position 39 and 40. As shown in Figure 4d, incorporation of abasic site results reduces in fluorescence with the stronger effect occurring at position 39.

Figure: Importance of binding site nucleobases and the P1 stem for OTB-SO₃ fluorescence activation (RNA concentration 3 μM and OTB-SO₃ 6 μM) T and L denote transcribed and ligated aptamers, respectively. The error bars indicate mean and standard deviations from three consecutive measurements.

6. The most interesting part of this paper is that this is the first fluorophore-binding and -enhancing aptamer that works without a G-quartet. The authors should consider putting this into the title of their paper. It is profound given that eukaryotic cellular conditions unfold G-quartets, and it is a non-obvious and non-predictable results because G-quartets in aptamers are not always easily predicted.

Based on this comment and comment 6 from reviewer 3, we have revised the title as follows:

Structural basis for binding and activation of fluorogenic dyes by an RNA aptamer that lacks a G-quadruplex motif.

7. The similarity of the ligand-bound and ligand-free structures suggests a role for minimal entropy loss in ligand binding, especially given the paucity of interactions with the ligand (e.g. no interactions with the benzoxazole ring). It was surprising that this feature wasn't discussed.

We thank reviewer for pointing this out. We have added the following text in the discussion section of the revised manuscript to emphasize this point.

Similar to previous observations for the Spinach aptamer, the DIR2s aptamer retains its global architecture in the absence of ligand and undergoes only a local structure perturbation to accommodate the fluorophore. The ligand binding site is largely preorganized and requires only a modest rotation at the glycosidic bond of A40, suggesting that ligand binding imparts a minimal entropic penalty.

8. The authors obtain their crystals in 1.8 M lithium sulfate. (a) This is given as LiSO_4 in the Materials and Methods but should be Li_2SO_4 .

We thank reviewer for catching this typo, we corrected this in Materials and Methods section.

(b) 3.6 M Li^+ would preclude G-quadruplex formation. Does the structure change when K^+ or Na^+ , which favor G-quadruplex formation, are added? This could be assayed by comparing fluorescence in these different monovalent salts at more reasonable (100 mM) salt concentration.

This experiment was done in the original paper *J. Am. Chem. Soc.* 139, 9001-9009 (2017) by Armitage group, in Figure 5B, where they have tested the effect of K^+ , Na^+ and Li^+ on fluorescence activation of DIR and OTB dyes and showed that fluorescence was not affected by presence of any of these monovalent ions. We have noted this important point in the revised manuscript.

Figure: Effect of monovalent ions on fluorescence activation of OTB-SO₃ and DIR dyes by DIR2s aptamers. Figure adapted from J. Am. Chem. Soc. 139, 9001-9009 (2017).

Minor comments

I found it confusing and somewhat off-putting to find “DIR2s” in the title and opening line of the Abstract without any decoding or explanation. Later we are provided that this is a second-generation binder for a fluorophore called DIR, although I still don’t know where the ‘s’ comes from. Please explain earlier.

We apologize for this oversight. DIR2s is an aptamer that emerged from a second selection against DIR using a different randomized library compared to the first selection. We have included some brief explanation for this in the abstract and in the manuscript. We have also removed ‘DIR2s’ from the title.

The description of the Fab interactions on p4/5 is fairly ordinary and could be moved to the SI.

We agree with the reviewer. We have moved the description of the RNA-Fab interactions to the Supplementary Information.

Chemical structures of DIR-SO₃ and OTB-SO₃ are dually shown in Fig. 1 and Supplementary Fig. 1. The authors should delete the structure on Supplementary Fig. 1 to prevent confusion.

We agree and have deleted the structures of dyes from Supplementary Figure 1.

The figure call outs in the ms goes straight from Fig1 to Fig 3.

Thank you for pointing it out, we have fixed the chronological order issue.

Fig. 4d, what does the error bars mean?

In the revised manuscript, we have added a description of the meaning of the error bars to the figure legend.

Supplementary Fig. 4 and Supplementary Fig. 9 are not used in the main text.

The revised manuscript now cites both Supplementary Figure 4 and Supplementary Figure 9 (Supplementary Figure 13 in revised manuscript). We thank the reviewer for pointing this out.

Supplementary Fig. 9. Y axis is weird. Why do they need to change the scale change between 1×10^7 and 0? In other words, just report as 0.5×10^7 .

We have changed the Y axis scale in Supplementary Figure 9 (Supplementary Figure 13 in revised manuscript) as suggested by the reviewer.

Also Fig. 4d shows that the fluorescence for 0 bp P1-aptamer is ~20% smaller than that for 5 bp P1-aptamer. Despite the same condition, why the fluorescence for these aptamers has similar values in Supplementary Fig. 9?

The fluorescence values reported in figure 4d were measured at room temperature (approximately 22 °C), which shows that the fluorescence for 0 bp P1-aptamer is ~20% smaller than that for 5 bp P1-aptamer. In Supplementary Figure 9, the fluorescence at 22 °C for 0 bp P1 aptamer is ~23% smaller than that for 5 bp P1-aptamer, which is consistent with Figure 4d. Perhaps the reviewer is referring to a lower temperature in Supplementary Figure 9 (Supplementary Figure 13 in revised manuscript), where the values for the 0 and 5 bp P1 aptamers are more similar.

Fig 1b. Capitalization of the dyes is uneven.

Thank you for catching this. We have now made the capitalization of the dyes consistent throughout the manuscript and Supplementary Information.

At one point the authors refer to the G as having an amidinium group and another as a guanidinium group. Either can be used, although I prefer the former, but the authors should be consistent.

We agree; we have changed the guanidinium to an amidinium.

Reviewer #2 (Remarks to the Author):

The authors report crystal structures of an in vitro evolved RNA aptamer that binds two different fluorescent compounds, dimethylindole red and oxazole thiazole blue, as well as the structure of the RNA in the absence of either ligand. The experimental crystallographic results all appear to be quite sound, and represented accurately. However, Figure 3b does not give us a good view of the quality of the electron density map and difference Fourier.

We thank the reviewer for the positive comments and appreciate the time and efforts taken to review the manuscript.

To give the better view of the overall density maps, we have provided the overall $|F_o| - |F_c|$ electron density maps of both the ligand-bound and ligand-free forms of the aptamer in Supplementary Figure 15 and a figure (Supplementary Figure 16) showing the B-factors corresponding to each region.

The three RNA crystal structures are quite similar to one another; the main difference in the apo structure is likely a minor artifact due to crystal contact interactions. Although several previous RNA aptamer structures that bind to fluorescent compounds have been previously determined, this RNA is unique in that it does not possess a G quadruplex platform upon which the fluorescent compound stacks. Instead, this RNA possesses a base triple (A15-A41-G39) as a stacking platform, which appears to be the primary claim to novelty of the paper, although the absence of the potential for formation of a G tetraplex was previously apparent from the sequence and secondary structure of the aptamer.

This claim bothers me a bit, because that triple is part of a coplanar array of four purines (A15-A41-G39-A14), which is clearly analogous to (but not identical to) a (purine) G-tetrad. In other words, it looks like the authors, in their zeal to make the structure sound unique, have overlooked or de-emphasized what is to me the most interesting result -- that there is more than one way to make a purine tetrad that supports fluorescent ligand binding. The paper would be much more interesting if this aspect was elaborated upon. What is doing on, electronically, with respect to the purine binding platform structures. Presumably, it has to do with pi-orbital conjugation. Perhaps it is beyond the scope of the current manuscript, but this should be elucidated, perhaps in collaboration with a computational quantum chemist.

We apologize for the misunderstanding regarding the presence of a tetrad. The structure does not contain an A15-A41-G39-A14 tetrad. The platform for ligand stacking involves a triad that includes A15, A41, and G39. Beneath this triad there is a second triad consisting of A14, U42, U38. To clarify this confusion, we have now changed the view of the figure 2a and made the color of the nucleotides in figures 2b and 2c consistent with figure 2a. See below for the revised Figure 2.

We appreciate the reviewer's comment regarding the electronic factors that dictate the purine binding platform structures. We have begun to work with a computational chemist to try to understand the electronics of the ligand-receptor interaction.

Reviewer #3 (Remarks to the Author):

In this manuscript the authors solved the crystal structure of DIR2s aptamer in the apo and holo form (bound to the blue fluorescent oxazole thiazole blue, OTB-SO₃, fluorophore). The DIR2s aptamer has been previously shown to be a promiscuous RNA aptamer, binding to both blue fluorescent OTB-SO₃ and red fluorescent dimethylindole red (DIR-SO₃) and has been utilized to image epidermal growth factor receptor (EGFR) when fused to an EGFP-binding RNA aptamer. Even though the authors compared DIR2s to Spinach and Mango which have been demonstrated to be valuable tools for imaging RNAs in live cells, the usefulness of the DIR2s in RNA imaging has not been proven yet, which in my opinion reduces the significance of this work for a broader community. While the provided structure is certainly interesting, a structure of the red dye in complex with aptamer would be more relevant to potential users, as blue fluorescence is much more difficult to implement in biological systems. The stoichiometric mismatch between crystallographic (2:2) and solution experiments (1:1) concerns me. Specific comments are provided below:

We thank reviewer 3 for the valuable comments and suggestions. We did set up crystallization trials of DIR2s aptamer with the red dye and obtained only crystals that diffracted poorly. At this stage, we do not know the exact reason for the weak diffraction of the crystals. However, we do expect that DIR-SO₃ would not support the intermolecular crystal packing interaction mediated by OTB-SO₃ (Fig. 3d) due to the longer linker in DIR-SO₃ that separates the heterocycles. We are continuing our attempts to optimize the crystallization conditions.

As the reviewer points out, the DIR2s aptamer has not been used for RNA imaging inside the cells, but it has been useful for extracellular imaging of a cell surface receptor and its internalization into cells. As described below, we believe that the structure will inform the design of cell permeable ligands that can bind to the aptamer. We think the structure has significance as one of the only fluorescence activation aptamers that doesn't use a G-quadruplex motif.

Our responses to the reviewer's specific comments follow.

1. It is not clear if replacing the UUCG tetraloop with GAAACAC sequence and binding of the Fab to the modified-aptamer has any effect on the binding affinity of the aptamer to the fluorophores. The authors only compared the fluorescence values, presumably in the

presence of very high concentrations of the aptamer or dye (Figure S1c). K_d values should be reported here to really understand the influences of those changes.

The revised manuscript now reports the K_d values also for the GAAACAC-modified aptamer. The affinities of the GAAACAC mutant and the UUCG WT aptamers with OTB-SO₃ (blue dye) within the experimental error (K_d , UUCG aptamer = ~700 nM according to J. Am. Chem. Soc. 139, 9001-9009, 2017 and K_d , GAAACAC aptamer = ~700 nM measured here) of one another. Consistent with the observed similarity, in the crystal structure the GAAACAC loop directs Fab BL3-6 away from the dye binding site.

2. In Figure 4a, why is there a data point with a negative fluorescence intensity?

We thank the reviewer for catching this error. We reproduced the experiments under similar conditions and this error has been corrected in the revised manuscript.

3. In Figure 4a, authors show that when 10 μ M of the fluorophore is titrated with up to 1.5 μ M of DIR2s, they obtain a linear increase in fluorescence with slope of 1.08 and conclude that this shows 1:1 binding. First of all, the slope in this graph does not equal to 1. Fluorescence intensity has arbitrary units and can be normalized to any value. This would change the calculated slope. Therefore, the value of the slope does not mean much. In

addition, if the affinity for dimer formation (i.e., the 2:2 dye: aptamer complex) is very strong (i.e., $K_d < 2$ nM), one would still observe a linear increase in this experiment. A more meaningful experiment is to do the same experiment with higher concentration of the DIR2s (for instance up to 30 μM). If you reach the plateau when 10 μM of DIR2s added, it means 1:1 binding (or 2:2). Similarly, if you reach the plateau when 20 μM of DIR2s added, it means 1:2 binding, etc.

We agree that the slope is not meaningful here; our intent was to show the increase in fluorescence was linear over the aptamer concentration range tested, expecting that if dimerization occurred in this concentration regime, we would see deviation from linearity. However, as pointed out by the reviewer, if the affinity of the dimer was very strong, our strategy would not be effective. We have performed the experiment suggested by the reviewer and the full graph is included below and in the revised manuscript as Figure 4a and Supplementary Figure 8. As predicted by the reviewer, we observed a plateau at ~ 10 μM DIR2s aptamer when titrated against 10 μM of the blue dye, consistent with 1:1 (or 2:2) stoichiometry. We obtained analogous results for the red dye. In addition, we note that the Armitage group (J. Am. Chem. Soc. 139, 9001-9009, 2017) performed continuous variation experiments (described below) and concluded 1:1 (or 2:2) binding between the aptamer and the dye. These experiments together with new SEC data (see response to point 3 from Reviewer 1 and response to point 7 below) and the SAXS analysis are consistent with monomeric form of the aptamer in solution binding to ligand with 1:1 stoichiometry.

Figure: Fluorescence activation of OTB-SO₃ dye as a function of DIR2s RNA aptamer concentration. Fluorescence signal increases linearly ($R^2 = 0.996$) with aptamer concentration in the range of 0–2.5 μM (right) as well as a plateau observed at ~ 10 μM in the aptamer titration

(left) indicating, 1:1 RNA:fluorophore complex formation in solution. The concentration of OTB-SO₃ was 10 μM.

V. Stoichiometries Determined by Continuous Variations Experiments

Samples containing dye and aptamer in varying ratio but constant total concentration of 1 μM were prepared in binding buffer. Fluorescence spectra were recorded with appropriate excitation wavelengths for each dye. The maximum fluorescence is plotted versus dye mole fraction in Figure S6. In each case, the data show a clear maximum at mole fraction of 0.5, corresponding to a 1:1 stoichiometry.

Figure S6. Binding stoichiometry determination of DIR2s-Apt with four dyes. The total concentration of dye plus RNA is 1 μM.

Figure adapted from *J. Am. Chem. Soc.* 139, 9001-9009, 2017

4. In Figure 4c, authors show that the fluorescence of the DIR2s-OTB-SO₃ complex decreases upon addition of DIR dye, and conclude that both dyes bind to the same site. However, it is also possible that DIR dye binds to somewhere else, influencing the binding pocket of OTB-SO₃ and causing OTB-SO₃ to dissociate.

A similar concern was also raised by Reviewer 1 (see response to point 4). We agree that the observation that the two dyes compete with each other for binding to the aptamer is only consistent with them occupying the same site; it is not conclusive proof as these data do not rule out the alternative possibility that the dyes compete by binding to distinct sites on the RNA that are mutually exclusive for dye binding. We have stated this point in the revised manuscript.

In addition to the competition data several other observations bolster the likelihood that the OTB-SO₃ and DIR-SO₃ binding sites at least partially overlap: both dyes have the same response to mutations; both dyes contain a cationic heterocycle that bears a propyl sulfonate side chain that can be accommodated at the same position in the aptamer structure; the lack of a cap over the benzoxazole ring in the structure OTB-SO₃-DIR elicits the expectation that more than one type of dye can bind in this site.

Figure: Important nucleobases at the binding site and the P1 stem that contribute for DIR-SO₃ binding and fluorescence activation. RNA and DIR-SO₃ concentrations were 3 μ M and 6 μ M, respectively.

Figure: Important nucleobases at the binding site and the P1 stem that contribute for OTB-SO₃ binding and fluorescence activation. RNA and OTB-SO₃ concentrations were 3 μM and 6 μM, respectively.

5. Permeability of the dyes?

In the current form, these dyes are not cell permeable as they give very low background if added to cells in the absence of aptamer. They have been used with the aptamer only for imaging a cell surface receptor and its internalization., which conceivably could provide a mechanism to deliver the dyes into cells in some applications. Importantly, the structure provides insights into where permeabilizing or even organelle-targeting features might be appended onto the dyes without affecting aptamer binding.

6. Mismatch between title and content – the promiscuity aspect is only speculated about, not experimentally demonstrated. To demonstrate the structural basis for promiscuous binding one would need more ligand-bound structures than 1.

We have revised the title of the current manuscript as follows: Structural basis for binding and activation of fluorogenic dyes by an RNA aptamer that lacks a G-quadruplex motif.

7. If two dyes stack directly upon each other (as shown in Fig. 3c, d) this stacking will influence the photophysics. It might result in a quenched dimer or in an eximer, but it will most likely not have the same fluorescence properties as the monomeric aptamer-dye complex with the top face exposed to water. Recording fluorescence spectra of the crystals, and also of 1:1 mixture of aptamer and dye from very dilute to very concentrated should more convincingly resolve this dimer vs. monomer issue.

We think that photophysical studies of the aptamer will be insightful and important. Given that the SAXS analysis, new RNA-fluorophore titrations, and SEC experiments strongly suggest the monomer as the functional form, we plan to use RNA nanostructures to juxtapose the aptamers in a head to head orientation with the aim of facilitating formation of the dimeric form. It will be fascinating to perform photophysical studies on this construct in comparison to the monomer. One possible outcome, supported by literature (Di Fiori, N.; Meller, A. The Effect of dye-dye interactions on the spatial resolution of single-molecule FRET measurements in nucleic acids. *Biophys. J.* 2010, 98, 2265–2272; Zhegalova N. G., He S., Zhou H., Kim D. M., and Berezin M. Y., Minimization of self-quenching fluorescence on dyes conjugated to biomolecules with multiple labeling sites via asymmetrically charged NIR fluorophores, *Contrast Media Mol. Imaging*, 2014, 9,

355–362), is that stacking of the two dyes in the dimer complex would quench fluorescence.

Figure: Size exclusion chromatography (SEC) of DIR2s aptamer at concentrations ranging from 1 to 20 μM in complex with 2 equivalents of OTB-SO₃ dye for each concentration. RNA standards with molecular weight (MW) ~21 kDa (Red) and ~39 kDa (green) serve as the references for the monomeric (~18 kDa) and dimeric (~36 kDa) forms of the aptamer, respectively. Over the concentration range tested, DIR2s RNA elutes as a single peak slightly slower than 21 kDa reference (see vertical dotted line for guidance) expected for the monomeric form of the aptamer in the solution. All SEC experiments were carried out in a 50 mM Tris pH 7.4, 150 mM NaCl, 5 mM MgCl₂ buffer at 4 °C using a Superdex 200 10/300 GL column (GE healthcare).

8. Citations of fluorogenic aptamers are incomplete.

Thanks to the reviewer for this comment. We have now added more citations for several other fluoromodules listed below.

Lauhon CT, Szostak JW. RNA aptamers that bind flavin and nicotinamide redox cofactors. *J. Am. Chem. Soc.* **117**, 1246-1257 (1995).

Burgstaller P, Famulok M. Isolation of RNA aptamers for biological cofactors by in vitro selection. *Angewandte Chemie International Edition in English* **33**, 1084-1087 (1994).

Holeman LA, Robinson SL, Szostak JW, Wilson C. Isolation and characterization of fluorophore-binding RNA aptamers. *Fold Des.* **3**, 423-431 (1998).

Sunbul M, Jäschke A. Contact-Mediated Quenching for RNA Imaging in Bacteria with a Fluorophore-Binding Aptamer. *Angew. Chem. Int. Ed.* **52**, 13401-13404 (2013).

Sunbul M, Jäschke A. SRB-2: a promiscuous rainbow aptamer for live-cell RNA imaging. *Nucleic Acids Res.*, (2018).

Sando S, Narita A, Hayami M, Aoyama Y. Transcription monitoring using fused RNA with a dye-binding light-up aptamer as a tag: a blue fluorescent RNA. *Chem. Commun.*, 3858-3860 (2008).

Murata A, Sato S-i, Kawazoe Y, Uesugi M. Small-molecule fluorescent probes for specific RNA targets. *Chem. Commun.* **47**, 4712-4714 (2011).

Sato Si, Watanabe M, Katsuda Y, Murata A, Wang DO, Uesugi M. Live-Cell Imaging of Endogenous mRNAs with a Small Molecule. *Angew. Chem. Int. Ed.* **54**, 1855-1858 (2015).

Arora A, Sunbul M, Jäschke A. Dual-colour imaging of RNAs using quencher-and fluorophore-binding aptamers. *Nucleic Acids Res.* **43**, e144-e144 (2015).

Brasemann E, *et al.* A multicolor riboswitch-based platform for imaging of RNA in live mammalian cells. *Nat. Chem. Biol.*, (2018).

9. Last sentence of abstract inaccurate. Correct statement would be “unique among fluorogenic RNAs with known 3D structure”.

We have corrected this in the revised manuscript.

10. Minor textual inconsistencies (head-to-tail vs. end-to-end, p. 9)

We appreciate the reviewer catching these inconsistencies. The revised manuscript uses consistent terminology.

Reviewers' Comments:

Reviewer #1:

Remarks to the Author:

The authors have addressed all of my concerns. They were very thorough and conducting extensive new studies. I find the paper now acceptable for publication in Nature Communications.

Reviewer #2:

Remarks to the Author:

The authors have adequately addressed my concerns, and appear to have also done so for the other reviewers.

In the future, allowing a reviewer access to the diffraction data and PDB coordinates (and possibly pre-calculated density maps) would be very helpful.

Reviewer #3:

Remarks to the Author:

The authors have responded constructively to all points raised in my previous review. I am satisfied with the changes in the manuscript and with the direct answers in the rebuttal letter.